# Near-infrared to ultra-violet frequency conversion in chalcogenide metasurfaces

Jiannan Gao[1], Maria Antonietta Vincenti[2], Jesse Frantz [3], Anthony Clabeau[4], Xingdu Qiao[5], Liang Feng[6], Michael Scalora[7] & Natalia M. Litchinitser [1✉]

Chalcogenide photonics offers unique solutions for a broad range of applications from mid-infrared sensing to integrated, ultrafast, ultrahigh-bandwidth signal processing. However, to date its usage has been limited to the infrared part of the electromagnetic spectrum, thus avoiding ultraviolet and visible ranges due to absorption of chalcogenide glasses. Here, we experimentally demonstrate and report near-infrared to ultraviolet frequency conversion in an $As_2S_3$-based metasurface, enabled by a phase locking mechanism between the pump and the inhomogeneous portion of the third harmonic signal. Due to the phase locking, the inhomogeneous component co-propagates with the pump pulse and encounters the same effective dispersion as the infrared pump, and thus experiences little or no absorption, consequently opening previously unexploited spectral range for chalcogenide glass science and applications, despite the presence of strong material absorption in this range.

[1] Department of Electrical and Computer Engineering, Duke University, Durham, NC 27708, USA. [2] Department of Information Engineering, University of Brescia, Via Branze 38, 25123 Brescia, Italy. [3] US Naval Research Laboratory, 4555 Overlook Ave., SW, Washington, DC 20375, USA. [4] University Research Foundation, 6411 Ivy Ln. 110, Greenbelt, MD 20770, USA. [5] Department of Electrical and Systems Engineering, University of Pennsylvania, Philadelphia, PA 19104, USA. [6] Department of Materials Science and Engineering, University of Pennsylvania, Philadelphia, PA 19104, USA. [7] Aviation and Missile Center, US Army CCDC, Redstone Arsenal, AL 35898-5000, USA. ✉email: natalia.litchinitser@duke.edu

Chalcogenide glasses have emerged as one of the superior material systems for infrared photonics. Their high linear refractive indices and strong Kerr nonlinearity have already made them a promising platform for on-chip signal processing, such as optical switching, wavelength conversion and regeneration, molecular fingerprinting, environmental monitoring, sensing applications, and astronomy[1–8]. In addition, the photo-modification properties of chalcogenide glass make it suitable for direct laser-writing techniques of ultra-small photonic components that can be easily written into chalcogenide glass film in a single step[9]. While infrared applications of chalcogenides are flourishing, their use in the ultraviolet (UV) part of the electromagnetic spectrum was thought to be hindered by the excessively large absorption. It is noteworthy that numerous modern applications, including underwater communications, environmental monitoring, and biomedical imaging, require UV light sources[10–13]. Compact, integrated sources of UV radiation compatible with existing photonic platforms are therefore in significant demand.

Here we demonstrate that, as counterintuitive as it may appear, highly nonlinear (NL) chalcogenide glass, transparent in the near-infrared (NIR), can also be used to generate third harmonic (TH) frequencies in the UV part of the spectrum, despite the presence of strong material absorption in this range. We consider a photonic metasurface consisting of arsenic trisulfide ($As_2S_3$) nanowires deposited on a glass substrate, as shown in Fig. 1. A 300-nm-thick $As_2S_3$ film was deposited on a glass substrate by thermal evaporation, and then nanopatterned using electron-beam lithography with ZEP520A resist served as the mask, followed by subsequent reactive ion etching (see Methods). The dimensions of the nanowire were optimized to enhance third harmonic generation (THG) efficiency with a pump tuned to 1064 nm. The period of the nanostructure is $p = 625$ nm, the width and height of the nanowire are $w_x = 430$ nm and $h = 300$ nm, respectively. The refractive index and absorption coefficient of 300 nm $As_2S_3$ film are measured using a spectroscopic ellipsometer, as shown in Fig. 2B, indicating that $As_2S_3$ is highly dispersive and strongly

absorbing at wavelengths below 500 nm, where our TH wavelength (354 nm) is tuned.

## Results

**Theoretical foundation.** Let's consider a pump pulse transmitted across an interface between a linear and a nonlinear (NL) medium. In the absence of phase matching, there are always three generated harmonic components—one is reflected back into the linear medium, and the remaining two are transmitted. If phase matching is satisfied, then the two transmitted components are degenerate. The detailed theoretical description of this process based on the solution of Maxwell's equations, including homogeneous (HOM) and inhomogeneous (INHOM) components has been developed in refs. [14,15]. In the absence of absorption, the HOM-TH component propagates with the group velocity corresponding to material dispersion at the TH wavelength: $k_{HOM}(3\omega) = k_0(3\omega)n(3\omega)$. Here, $k_0(3\omega) = 3\omega/c$ is the wavenumber for the TH in a vacuum, and $n(3\omega)$ is the refractive index corresponding to the TH wavelength. However, the INHOM-TH, referred to as a phase-locked (PL) component[16,17], is trapped by the pump pulse and co-propagates with the wavenumber given by $k_{INHOM} = 3k_0(\omega)n(\omega)$, where $k_0(\omega)$ is the wavenumber of the FF in a vacuum, and $n(\omega)$ is the refractive index corresponding to the pump wavelength[14]. In particular, in ref. [16] it was shown that the PL components form only in the presence of either an interface or feedback. Therefore, while the harmonic fields are captured by the pump and propagate with it under anomalous dispersive conditions, in a bulk medium there is no energy exchange between the fundamental frequency (FF) and its PL components regardless of material thickness, as experimentally verified in refs. [18–21].

We now consider the case where the FF wave belongs to the transparency range of the material, while the TH frequency falls into the absorption range. Since the PL solution propagates with the phase and group velocities corresponding to the FF wave, the Kramers–Kronig relations suggest that the harmonic signal experiences the imaginary part of the refractive index corresponding to the FF wave. This behavior is characteristic of a driven oscillator, where the properties of the INHOM solution are entirely defined by the driving force. As a result, the propagation of the PL component is not affected by the material dispersion and absorption of the material at its wavelength corresponding to the opaque part of the spectrum, as long as the FF falls in the transparent or partially transparent spectral range. As mentioned above, the thickness of the medium or the nanostructure plays no role, unless it is part of a resonant cavity or a nanostructure. The PL component thus survives and resonates if the pump resonates, and is absorbed only if the pump is absorbed. This phenomenon has also been predicted and experimentally demonstrated in semiconductor substrates such as GaP and GaAs[19–22] and cavities[23,24].

Next, we discuss our experimental measurements of the transmitted component of the THG as a function of frequency for an $As_2S_3$ metasurface with the parameters defined in Fig. 1. In parallel, we performed numerical simulations based on a time-domain theoretical model that couples material equations of motion to the macroscopic Maxwell's equations, and compared them with the experimental results. Typical theoretical descriptions of NL optical interactions in nanostructures most commonly rely on assigning a phenomenological, effective volume nonlinear susceptibility $\chi^{(3)}$ to describe THG, however, without specifying its origin and with the neglect of NL dispersion. We use a hydrodynamic model that preserves linear and NL material dispersions, allowing us to account for surface, magnetic, and bulk nonlinearities (described in Supplementary Note 1)[25,26]. Figure 2A shows the scanning electron microscopy

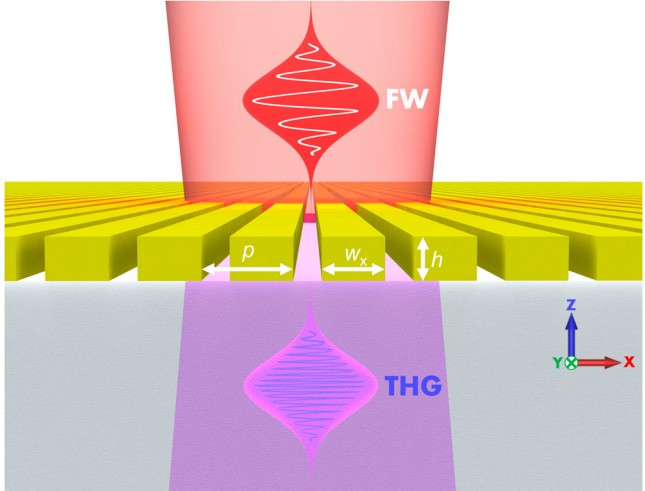

**Fig. 1 Third harmonic generation in the ultra-violet spectral range of $As_2S_3$.** An arsenic trisulfide nanowire-based metasurface on a silica substrate, enabling the desired enhancement of third harmonic generation. The $As_2S_3$ on $SiO_2$ 400 μm × 400 μm metasurface with a period $p = 625$ nm, nanowire width $w_x = 430$ nm and height $h = 300$ nm is pumped by the near-infrared 100 fs pulses from a tunable ultrafast laser system consisting of a 1 kHz Ti:sapphire laser and an optical parametric amplifiers (OPA), polarized along x-direction.

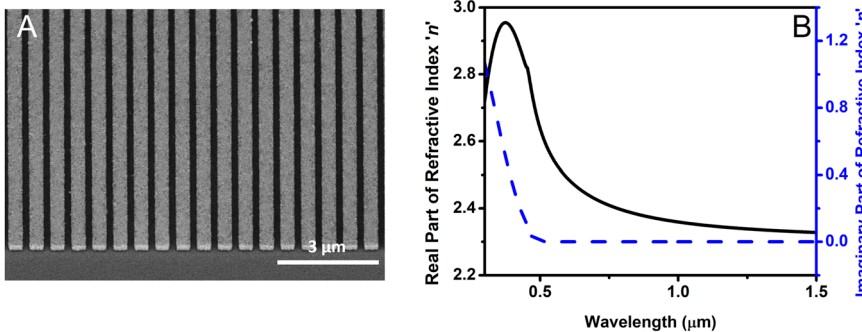

**Fig. 2 The designed metasurface and refractive index of As₂S₃. A** The 30-degree tilted SEM image of the patterned As$_2$S$_3$ structure on a glass substrate. **B** The real and imaginary part of the refractive index of a 300-nm-thick As$_2$S$_3$ thin film was measured using spectroscopic ellipsometry (Cauchy model). The absorption coefficient significantly increases for wavelengths below 500 nm.

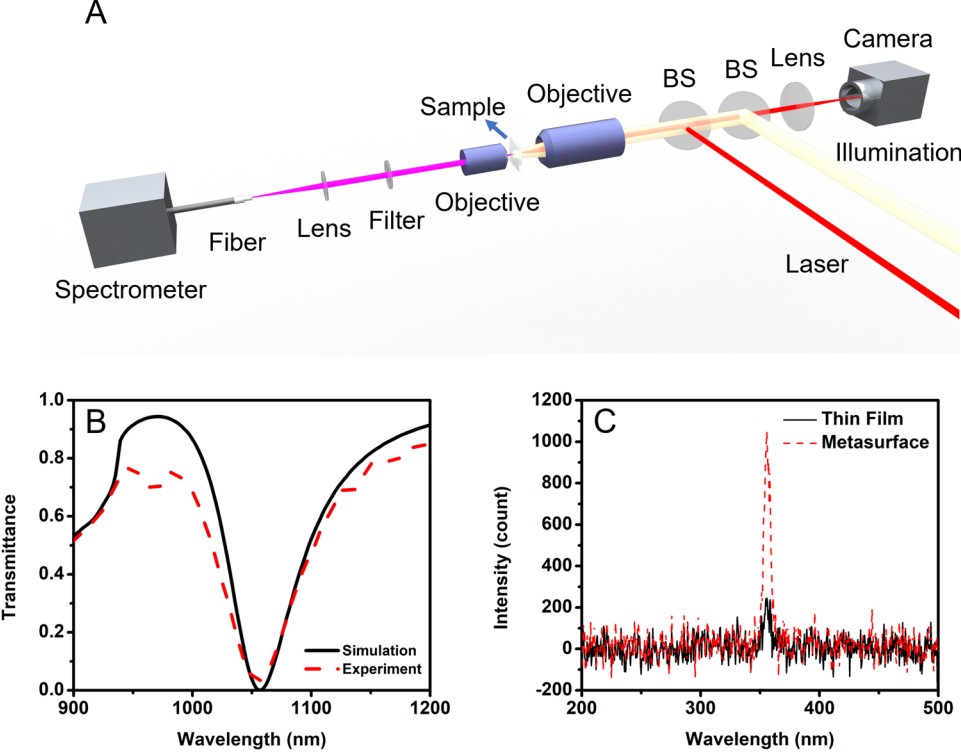

**Fig. 3 Third harmonic generation measurements results. A** Schematic illustration of the experimental setup. **B** Measured transmittance of the fabricated sample. The transmission dip appearing at around 1054 nm is in agreement with numerically simulated transmittance and experimental results. **C** Third harmonic intensity was measured for the uniform thin film and for nanowire-based metasurface. The intensity of the THG from the metasurface is about 5.5 times larger than that from the reference 300-nm-thick As$_2$S$_3$ thin film.

(SEM) image of the As$_2$S$_3$ pattern. The refractive index of the 300 nm As$_2$S$_3$ film, measured by ellipsometry and used in our design and numerical simulations, is shown in Fig. 2B. The nanopatterning step includes standard electron-beam lithography (EBL) with ZEP520A resist used as the mask and subsequent reactive ion etching step (see Methods).

### Experimental results

Figure 3A shows the diagram of the experimental setup. A Ti:sapphire pulsed laser (100 fs pulse width, 1 kHz repetition rate) light was routed into an OPA, where the NIR pulses were generated and redirected to the propagation path by a beam splitter (BS). A 4f system, consisting of an objective and an achromatic lens, was used for sample alignment and imaging. We used an infinity-corrected objective lens with an NA of 0.5 to focus the light on the pattern. The TH signal was collected by another 4f

system behind the sample and coupled into a spectrometer through a multimode fiber (400 μm core, 0.5 NA). Measured transmittance of the fabricated sample at wavelengths around the pump wavelength is shown in Fig. 3B. Figure 3C shows the experimental measurements of THG intensity from the metasurface and the unpatterned 300 nm As$_2$S$_3$ thin film. The fundamental wavelength is centered at a wavelength of 1064 nm, the beam waist is 20 μm, and the peak intensity is 1.4 GW cm$^{-2}$. These results clearly show that the THG from the metasurface is about 5.5 times greater than that from the uniform 300 nm thin film. The peak intensity of 1.4 GW cm$^{-2}$ used here is significantly lower than the surface damage threshold for As$_2$S$_3$ glass, previously measured by Stegeman et al., of 8.4 GW cm$^{-2}$ at a wavelength of 1064 nm[27]. We note, however, that As$_2$S$_3$ exhibits a host of photoinduced effects at intensities below the damage threshold, and this may necessitate maintaining an

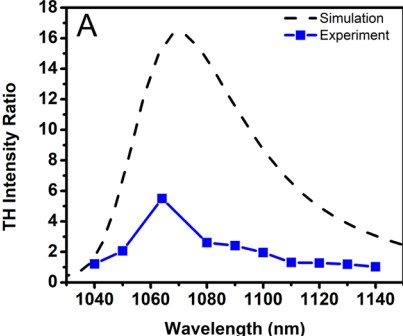
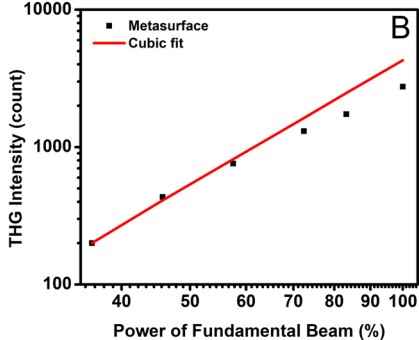

**Fig. 4 THG from different fundamental wavelengths and different pump powers. A** The simulated and experimental results of the THG enhancement for the metasurface as compared to that of the reference thin film. **B** The THG power dependence of the metasurface as a function of the power of the fundamental beam.

intensity lower than the laser-induced damage threshold. In addition, considering the electric field enhancement provided by the metasurface at resonance, the damage threshold of the metasurface will be lower than that of pure $As_2S_3$ thin film. It is likely that the peak pump intensity can be increased by replacing $As_2S_3$ with another chalcogenide glass with a composition optimized to maximize the threshold for the onset of damage (e.g., $Ge_{20}As_{20}Se_{60}$)[28] and by passivating the surface with a thin layer of alumina[29].

As the pump wavelength was swept from 1040 to 1140 nm, the enhancement of THG of a metasurface as compared to that of the reference thin film reaches its maximum at about 1064 nm, and then gradually decreases, which is in good agreement with the calculated results (Fig. 4A). While the maximum experimentally measured ratio of the TH intensity generated in the nanowire structure to that generated in the uniform thin film is smaller than the theoretically predicted ratio (see Supplementary Fig. 4), the following factors are likely to cause this discrepancy: (i) variations in the widths of the nanowires due to fabrication inaccuracies are expected to broaden the width of the resonance and decrease THG efficiency; (ii) the fundamental beam might not be perfectly normal to the metasurface due to the objective used. The THG power dependence is shown in the log scale in Fig. 4B, which proves that the THG power is proportional to the third power of the intensity of the fundamental beam. Note that when the power increases to 100%, the trend of the curve does not follow the cubic fit strictly. This difference can be attributed to the refractive index of the $As_2S_3$ changing as a function of the input light intensity due to its large NL Kerr response, which triggers a shift of the resonance.

Here we clarify how the diffraction orders at both pump and harmonic frequencies have been collected and/or discarded. Our chalcogenide metasurface has a periodicity $p = 625$ nm, and it is sandwiched between air and silica. For simplicity, we have assumed silica is dispersionless with refractive index $n = 1.5$ at both fundamental and TH frequencies.

Assuming normal incidence, the diffracted waves obey the following grating equation:

$$p\sin\theta_m = m\frac{\lambda}{n} \qquad (1)$$

Where $p$ is the periodicity of the structure, $m$ is the diffraction order ($m = 0, \pm 1, \pm 2, ..$), $\lambda$ is the wavelength under investigation, and $n$ is the refractive index of the medium adjacent to the periodic structure. We will therefore have a set of diffraction angles associated with reflected waves (airside), and the second set of angles associated with the transmitted waves (silica side).

On the airside ($n = 1$), the collectable diffraction orders are $\theta_0 = 0°$ at the pump wavelength $\lambda = 1064$ nm and $\theta_0 = 0°$,

$\theta_{\pm 1} = 34.5°$ at TH wavelength $\lambda = 354$ nm. On the silica side ($n = 1.5$), the collectable diffraction orders are $\theta_0 = 0°$ at the pump wavelength $\lambda = 1064$ nm and $\theta_0 = 0°$, $\theta_{\pm 1} = 39.8°$, $\theta_{\pm 2} = 49°$ at TH wavelength $\lambda = 354$ nm. In our experiment, the transmitted TH signal was coupled into a multimode fiber with $NA = 0.5$, which translates into a collection angle of 30° since the fiber is placed in the air ($n = 1$). This implies that our experimentally collected TH signal consists only of the zero-diffraction order, and that higher-order modes are discarded. When comparing theory and experiment we, therefore, discarded higher-order modes in the calculation of the TH conversion efficiency and reported only the conversion efficiency of the signal we could collect. Similar considerations were done when calculating the linear spectra of Figs. 3B and S3B: all higher-order modes that would fall outside the collection angle of the multimode optical fiber ($\theta > 30°$) were discarded in the calculated spectra to ensure a good match with the experimental conditions.

In summary, we theoretically predicted and experimentally demonstrated the possibility of THG in the opaque spectral range of $As_2S_3$ glass, enabled by phase-locking between the FF and TH waves, largely overlooked in the previous studies of NL light–matter interactions in chalcogenide glasses. TH generation is significantly enhanced by the excitation of a broad Mie resonance near 1050 nm (see Supplementary Fig. 3). Several solutions may be adopted to further enhance the NL process in the opaque regime of $As_2S_3$. While, in fact, we exploit a Mie resonance of the single nanowire to improve TH conversion efficiency, one may design the metasurface to support either guided-mode resonances or quasi-bound states in the continuum to further boost pump localization. Finally, by cascading additional metasurface layers, one could create a three-dimensional nanostructure that exhibits the typical spectrum of a photonic crystal, whose band-edges are well known to support strongly localized modes that can further enhance the NL coupling between the fields. However, the overarching effects that is at play in all types of possible geometries is the resonant aspect of the third-order NL coefficient.

## Methods

**Samples preparation.** The 300 nm $As_2S_3$ film was deposited on top of a glass substrate with thermal deposition in a Lesker PVD 75 deposition system equipped with a low-temperature evaporation source. The linear refractive index of the $As_2S_3$ was measured using a spectroscopic ellipsometer (VASE, J. A. Woollam). After cleaning the sample with acetone, isopropanol, and nitrogen, ZEP520A was spun to form a 120-nm-thick layer and baked for 1 min at 180 °C. A 15-nm-thick gold layer was sputtered on top of ZEP520A as the conductive layer for EBL. The pattern was exposed using an Elionix ELS-7500 EX E-Beam Lithography System and then developed in ZED-N50 developer after wet etching away the gold layer. The pattern was transferred to the $As_2S_3$ layer by standard reactive ion etching (Oxford Cobra). Finally, the ZEP520A mask was removed with 1165 Stripper (NMP). The SEM image of the pattern was obtained using an Apreo S system made by Thermo Fisher Scientific.

**Linear measurements**. A stabilized fiber-coupled light (SLS201, Thorlab) was used as a source and a wide range optical spectrum analyzer (AQ6374, Yokogawa) was used for spectral measurements. In addition, Thorlabs polarizer was used to ensure linear polarization of the incident on the sample beam. The transmission spectrum was calculated by dividing the power transmitted through the metasurface by that transmitted through free space.

**Nonlinear measurements**. A Ti:sapphire laser (100 fs output pulse width, 1 kHz repetition rate, Coherent Libra system) and an ultrafast optical parametric amplifier (TOPAS-C) was used as the tunable light source covering a range of 260 to 2600 nm for the center wavelength. The incoming laser light was focused by a 20X Mitutoyo Plan Apo NIR Infinity Corrected Objective and the transmitted FF and TH were collected with an AO 40x Long Working Distance LWD Plan Achro Objective with an NA of 0.55. Then the light is attenuated by NIR Absorptive ND Filters (NENIR40) and focused by an N-BK7 lens with a focal length of 250 mm. Next, the FF was filtered, and the TH was coupled into a multimode fiber with an 0.50 NA, Ø400 µm core (FP400URT) that was connected to a Super Gamut UV-VIS-NIR Spectrometer (BaySpec, Inc). A manual filter wheel mount with neutral density filters was used to change the power of the FF beam.

## Data availability
The data that support the findings of this study are available from the corresponding author upon reasonable request.

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

## Acknowledgements
This work was supported by Office of Naval Research (ONR) awards N00014-19-1-2163, N00014-20-1-2558, the Army Research Laboratory Cooperative Agreement W911NF-20-2-0078, the NSF ECCS-1846766, and OMA-1936276 awards.

## Author contributions
M.S., N.M.L., and M.A.V. developed the idea of this study. M.S. and M.A.V. performed theoretical and numerical studies. N.M.L. and J.G. designed and performed experiments. J.F. and A.C. deposited and characterized chalcogenide thin films. J.G., X.Q., and L.F. performed nanofabrication of the samples. All authors contributed to writing the manuscript.

## Competing interests
The authors declare no competing interests.
