## [Peer Review File · Nature Communications]

Near-Infrared to Ultra-Violet Frequency Conversion in Chalcogenide MetasurfacesREVIEWER COMMENTS

Reviewer #1 (Remarks to the Author):

I thought the paper was interesting and builds off of a lot of the chalcogenide glass foundation already established. Please find my review points below.

- Since the structure is periodic and not subwavelength to the 354nm light, the harmonic light should demonstrate diffraction. Analysis demonstrating which orders were collected/lost would significantly benefit this work.
- Figure 3B should show the theoretical linear transmission spectra of the metasurface. It is referred to in the figure caption, but not shown.
- The paper would significantly benefit from damage threshold analysis of the metasurface.

Reviewer #2 (Remarks to the Author):

In the manuscript “near-infrared to ultra-violet frequency conversion in chalcogenide metasurfaces” the authors analyze, theoretically and experimentally, third harmonic generation from Chalcogenide-based metasurfaces and demonstrate enhanced nonlinear frequency conversion in the materials that are opaque at the third harmonic frequency. The authors attribute this counterintuitive phenomenon to phase-locking between fundamental and nonlinear waves. Since phase-locking phenomenon is not specific to the particular material platform, the novel nonlinear optical mechanism could be utilized in other material systems and at other frequencies. The work therefore opens a new direction in applications of chalcogenide (and other strongly nonlinear but lossy) materials and will be of interest to researchers in the fields of metamaterials, metasurfaces, photonics, and materials science. As such, the manuscript should be published in Nat Comm.

However, prior to publication, I suggest the authors address the following points directed primarily at clarifying the main conclusions of the manuscript:

1. I suggest the authors move the comparison between experimentally measured and theoretically calculated transmission to the main body of the manuscript from SI. Also, a photo in Fig.3a could use the labels or be replaced with flow-chart-type schematic of the experiment.
2. Currently, the authors use two fundamentally different approaches to assess nonlinear performance of their devices: conversion efficiency in theoretical studies and ratio of THG signal in experiments. Adding theoretical estimates of ratio of THG signal (possibly, in SI) will provide a fairer comparison between theory and experiment
3. I suggest that the authors comment on possible extensions of their work that would enable higher overall conversion efficiencies. Also, the authors should also comment on whether the phase-locking mechanism, demonstrated in this work, can be utilized for enhancing nonlinear response of thicker structures.

Reviewer #3 (Remarks to the Author):

In this manuscript, Gao et al reported, for the first time, the NIR to UV frequency conversion through a third harmonic generation (THG) in chalcogenide metasurfaces. Chalcogenide

photonics has recently attracted significant attention, but the material is typically of high absorption in visible and UV ranges, which limits its applications. Despite strong absorption, the authors demonstrated the THG can still be conveniently facilitated using chalcogenide metasurfaces, leading to a source for strong UV radiation.

The work is innovative and the experimental results are consistent with the theoretical model.

I would like to recommend publication after the authors address the following comments:

1. To make the main text self-sufficient, the authors should consider adding a section to clearly describe the underlying phase-locking (or phase-matching) mechanism. Some materials in the supplementary information can be moved to the main text. How was the metasurface designed to satisfy the phase-locking?
2. Related to Comment 1, what was the incident angle considered in the theoretical model? From the setup, the incident NIR pulse is focused onto the sample, which means the incidence covers a spectrum of spatial frequencies. Is this a problem?
3. Nonlinear effects can be enhanced if resonances happen at both fundamental and TH frequencies. Figure S5 shows the resonance-assisted light-matter interaction at the fundamental frequency. It is also important to see if strong light-matter interaction has been also designed for the TH light.
4. What does absorption coefficient k stand for in Fig. 2b? The imaginary part of the refractive index of the material? It should be clearly defined in the figure caption.
5. While Mie resonance was applied in the current work, it would be important to discuss other resonant photonic metasurfaces by which a large variety of nonlinear effects become possible.

REVIEWER COMMENTS

We thank the referees for their thoughtful responses and are happy to note the generally positive and constructive tone of the critiques. Below we provide detailed responses to all points raised.

Reviewer #1 (Remarks to the Author):

I thought the paper was interesting and builds off of a lot of the chalcogenide glass foundation already established. Please find my review points below.

- Since the structure is periodic and not subwavelength to the 354nm light, the harmonic light should demonstrate diffraction. Analysis demonstrating which orders were collected/lost would significantly benefit this work.

We would like to thank the reviewer for this valuable comment. Here and in the manuscript, we clarify how the diffraction orders at both pump and harmonic frequencies have been collected and/or discarded.

Our chalcogenide metasurface has a periodicity $p=625\text{nm}$, and it is sandwiched between air and silica. For simplicity, we have assumed silica is dispersionless with refractive index $n=1.5$ at both fundamental and third harmonic frequencies.

Assuming normal incidence, the diffracted waves obey to the following grating equation:

$$p \sin\theta_m = m \frac{\lambda}{n}$$

Where p is the periodicity of the structure ($p=625\text{nm}$), m is the diffraction order ($m = 0, \pm 1, \pm 2, \dots$), λ is the wavelength under investigation and n is the refractive index of the medium adjacent to the periodic structure. We will therefore have a set of diffraction angles associated with reflected waves (air side), and a second set of angles associated with the transmitted waves (silica side).

On the air side ($n=1$) the collectable diffraction orders are:

- At pump frequency $\lambda = 1064\text{nm}$: $\theta_0 = 0^\circ$
- At TH frequency $\lambda = 354\text{nm}$: $\theta_0 = 0^\circ$, $\theta_{\pm 1} = 34.5^\circ$.

On the silica side ($n=1.5$) the collectable diffraction orders are:

- At pump frequency $\lambda = 1064\text{nm}$: $\theta_0 = 0^\circ$
- At TH frequency $\lambda = 354\text{nm}$: $\theta_0 = 0^\circ$, $\theta_{\pm 1} = 39.8^\circ$, $\theta_{\pm 2} = 49^\circ$.

In our experiment, the transmitted TH signal was coupled into a multimode fiber with NA=0.50, which translates into a collection angle of 30° since the fiber is placed in air ($n=1$). This implies that our experimentally collected TH signal consists only of the zero-diffraction order, and that higher order modes are discarded.

When comparing theory and experiment we, therefore, discarded higher order modes in the calculation of the TH conversion efficiency and reported only the conversion efficiency of the signal we could collect (Figure S4 B in the manuscript).

Similar considerations were done when calculating the linear spectra of Figure 3B and S3 B: all higher order modes that would fall outside the collection angle of the multimode optical fiber ($\theta > 30^\circ$) were discarded in the calculated spectra to ensure a good match with the experimental measurements (Figure S3 A in the manuscript).

- Figure 3B should show the theoretical linear transmission spectra of the metasurface. It is referred to in the figure caption, but not shown.

Following the suggestion of the reviewer, here and in the revised manuscript, we added the theoretical linear transmission spectra in Figure 3B.

Figure 3B: Measured transmittance of the fabricated sample. The transmission dip appearing at around 1054 nm is in agreement with numerically simulated transmittance, and results from the Mie Resonance of the nanowires.

- The paper would significantly benefit from damage threshold analysis of the metasurface.

We really appreciate this valuable comment. In the modified manuscript we added the following paragraph that explains that our operating regime is well below the surface damage threshold for As_2S_3 .

The peak intensity of 1.4 GW/cm^2 used here is significantly lower than the surface damage threshold for As_2S_3 glass, previously measured by Stegeman, et al., of 8.4 GW/cm^2 at a wavelength of 1064 nm [R. Stegeman, et al., "Raman gain measurements and photo-induced transmission effects of germanium- and arsenic-based chalcogenide glasses," *Opt. Expr.*, v 14, pp. 11702-11708 (2006)]. We note, however, that As_2S_3 exhibits a host of photo-induced effects at intensities below the damage threshold, and this may necessitate maintaining an intensity lower than the laser-induced damage threshold. In addition, considering the electrical field enhancement provided by the metasurface at resonance, the damage threshold of the metasurface will be lower than the that of pure As_2S_3 thin film. It is likely that the peak pump intensity can be increased by replacing As_2S_3 with a chalcogenide glass with a composition optimized to maximize the threshold for the onset of damage (e.g. $\text{Ge}_{20}\text{As}_{20}\text{Se}_{60}$) [P. Němec, et al., "Photostability of pulsed laser deposited $\text{Ge}_x\text{As}_y\text{Se}_{100-x-y}$ amorphous thin films," *Opt. Express*, v 18, pp. 22944-22957 (2010)] and by passivating the surface with a thin layer of alumina [X. Gai, et al., "Progress in optical waveguides fabricated from chalcogenide glasses," *Opt. Express*, v 18, pp. 26635-26646 (2010)].

Reviewer #2 (Remarks to the Author):

In the manuscript “near-infrared to ultra-violet frequency conversion in chalcogenide metasurfaces” the authors analyze, theoretically and experimentally, third harmonic generation from Chalcogenide-based metasurfaces and demonstrate enhanced nonlinear frequency conversion in the materials that are opaque at the third harmonic frequency. The authors attribute this counterintuitive phenomenon to phase-locking between fundamental and nonlinear waves. Since phase-locking phenomenon is not specific to the particular material platform, the novel nonlinear optical mechanism could be utilized in other material systems and at other frequencies. The work therefore opens a new direction in applications of chalcogenide (and other strongly nonlinear but lossy) materials and will be of interest to researchers in the fields of metamaterials, metasurfaces, photonics, and materials science. As such, the manuscript should be published in Nat Comm.

However, prior to publication, I suggest the authors address the following points directed primarily at clarifying the main conclusions of the manuscript:

1. I suggest the authors move the comparison between experimentally measured and theoretically calculated transmission to the main body of the manuscript from SI. Also, a photo in Fig.3a could use the labels or be replaced with flow-chart-type schematic of the experiment.

We are grateful to the reviewer for his/her suggestions. Here and in the revised manuscript, we added the theoretical linear transmission spectra in Figure 3B and **redrew Fig.3A with labels.**

Figure 3B: Measured transmittance of the fabricated sample. The transmission dip appearing at around 1054 nm is in agreement with numerically simulated transmittance, and results from the Mie Resonance of the nanowires.

Figure 3A: Schematic illustration of the experimental setup.

2. Currently, the authors use two fundamentally different approaches to assess nonlinear performance of their devices: conversion efficiency in theoretical studies and ratio of THG signal in experiments. Adding theoretical estimates of ratio of THG signal (possibly, in SI) will provide a fairer comparison between theory and experiment.

Figure 4. A. The simulated and experimental results of the THG enhancement for the metasurface as compared to that of the reference thin film.

As suggested by the referee, we have replaced to the theoretical estimate of the ratio between the generated signal from the metasurface with respect to the signal generated by the unpatterned layer in Fig 4A in the new version of the manuscript. To avoid redundancy, we are now simply indicating in the text that the maximum conversion efficiency achieved at the peak is approximately 2.1×10^{-8} .

The modified text in the supplemental section now reads as follows:

“Figure S4 A shows the experimental nonlinear measurements, while Figure S4 B reports the theoretical enhancement expected for the TH signal from the metasurface with respect to TH signal from the unpatterned As₂S₃ layer. The peak of the TH enhancement in panel B corresponds to a conversion efficiency of $\sim 2.1 \cdot 10^{-8}$.”

3. I suggest that the authors comment on possible extensions of their work that would enable higher overall conversion efficiencies. Also, the authors should also comment on whether the phase-locking mechanism, demonstrated in this work, can be utilized for enhancing nonlinear response of thicker structures.

We would like to thank the reviewer for the valuable suggestions. We have added the following statement in the summary on page 10, and it reads as follows:

“Several solutions may be adopted to further enhance the nonlinear process in the opaque regime of As₂S₃. While, in fact, we exploit a Mie resonance of the single nanowire to improve TH conversion efficiency, one may design the metasurface to support either guided mode resonances or quasi-bound states in the continuum to further boost pump localization. Finally, by cascading additional metasurface layers, one could create a three-dimensional nanostructure that exhibits the typical spectrum of a photonic crystal, whose band-edges are well known to support strongly localized modes that can further enhance the nonlinear coupling between the fields. However, the overarching effects that is at play in all types of possible geometries is the resonant aspect of the third order nonlinear coefficient.”

Regarding phase locking and thicker structures, starting on page 3, under “theoretical foundations”, the text of the first paragraph has been modified as follows:

When a pump pulse crosses an interface between a linear and a nonlinear (NL) medium, *in the absence of phase matching* there are always three generated harmonic components – one is reflected back into the linear medium, and the remaining two are transmitted. If phase matching is satisfied, then the two transmitted components are degenerate.”

At the end of the first paragraph on page 4, we have added the following sentence to clarify some aspects relating to the propagating of the phase locked component:

“In particular, in reference [16] it was shown that the phase locked components form only in the presence of either an interface or feedback. Therefore, while the harmonic fields are captured by the pump and propagate with it under anomalous dispersive conditions, in a bulk medium there is no energy exchange between the fundamental and its phase

locked components regardless of material thickness, as experimentally verified in references [18-21].”

Beginning 6 lines from the bottom of page 4, the new text reads as follows:

“As a result, we conclude that the PL harmonic component propagates inside the material regardless of material dispersion and absorption at any harmonic wavelength that falls in the opaque region of the spectrum, as long as the FF wavelength falls in the transparent or partially transparent spectral range. As mentioned above, the thickness of the medium or the nanostructure plays no role, unless it is part of a resonant cavity or nanostructure. The PL component thus survives and resonates if the pump resonates, and is absorbed only if the pump is absorbed. This phenomenon has been experimentally demonstrated in bulk semiconductor substrates such as GaP and GaAs [19-22] and in cavity configurations [23-24].

Reviewer #3 (Remarks to the Author):

In this manuscript, Gao et al reported, for the first time, the NIR to UV frequency conversion through a third harmonic generation (THG) in chalcogenide metasurfaces. Chalcogenide photonics has recently attracted significant attention, but the material is typically of high absorption in visible and UV ranges, which limits its applications. Despite strong absorption, the authors demonstrated the THG can still be conveniently facilitated using chalcogenide metasurfaces, leading to a source for strong UV radiation. The work is innovative and the experimental results are consistent with the theoretical model.

I would like to recommend publication after the authors address the following comments:

1. To make the main text self-sufficient, the authors should consider adding a section to clearly describe the underlying phase-locking (or phase-matching) mechanism. Some materials in the supplementary information can be moved to the main text. How was the metasurface designed to satisfy the phase-locking?

Thanks for the valuable comments. As mentioned in the response to the second reviewer, under generic phase mismatched conditions typical of naturally occurring materials near an absorption resonance, the nonlinear medium displays two transmitted components created simultaneously at the medium entrance: One is almost immediately absorbed with a few tens of nanometers from the entrance interface, while the phase locked component is captured by the pump but does not exchange energy with it until another interface is crossed. If the structure is resonant, i.e. in the presence of closely spaced interfaces and feedback, then both pump and harmonic signals undergo multiple passes that cause: (1) the local pump field to be amplified, (2) quick absorption of the homogenous component, and (3) efficient energy exchange between fundamental and phase-locked harmonic signal through exploitation of the combination of large local fields and resonant nonlinearities. Experimental evidence in bulk [18-22] and cavities [23-24] confirms this dynamics.

We have contextualized the above statement and added it on page 4.

2. Related to Comment 1, what was the incident angle considered in the theoretical model? From the setup, the incident NIR pulse is focused onto the sample, which means the incidence covers a spectrum of spatial frequencies. Is this a problem?

In theoretical model we considered normal incidence and in the experiment the incident NIR pulse was focused onto sample. The focusing will produce some k-vectors with angles and not all k-vectors are perfectly normal to the metasurface, leading to the decrease of the THG efficiency compared with the theoretical model. As written in the manuscript, this is one of the factors that are likely to cause the discrepancy between

simulation and experimental results. For the TH collection, we have considered the effect of NA from fiber coupling into simulation. Here is the calculated comparison of THG with different NA. At the TH wavelength we have 3 diffraction orders in transmission but only the 0th and 1st order will be detected. That's why the signal lowers when we go from NA=1 to NA=0.5.

3. Nonlinear effects can be enhanced if resonances happen at both fundamental and TH frequencies. Figure S5 shows the resonance-assisted light-matter interaction at the fundamental frequency. It is also important to see if strong light-matter interaction has been also designed for the TH light.

Yes, the reviewer is correct and typically having a structure that resonates at both pump and harmonic frequency will help achieve a better nonlinear light matter interaction. However, since As_2S_3 exhibits very large absorption at 354nm, no resonant features can be observed at TH wavelength (see figure below). Moreover, because the TH signal here is due only to the phase-locked component, only the field enhancement at the pump frequency is critical to improve the TH conversion efficiency: the phase locked component resonates if the pump resonates, regardless of the dispersion at the harmonic frequency.

4. What does absorption coefficient k stand for in Fig. 2b? The imaginary part of the refractive index of the material? It should be clearly defined in the figure caption.

Sorry for this unclear description in Fig. 2B. We have changed the absorption coefficient k to imaginary part of the refractive index here and in the revised manuscript.

5. While Mie resonance was applied in the current work, it would be important to discuss other resonant photonic metasurfaces by which a large variety of nonlinear effects become possible.

We have added the following statement in the summary on page 10, and it reads as follows:

“Several solutions may be adopted to further enhance the nonlinear process in the opaque regime of As_2S_3 . While, in fact, we exploit a Mie resonance of the single nanowire to improve TH conversion efficiency, one may design the metasurface to support either guided mode resonances or quasi-bound states in the continuum to further boost pump localization. Finally, by cascading additional metasurface layers, one could create a three-dimensional nanostructure that exhibits the typical spectrum of a photonic crystal, whose band-edges are well known to support strongly localized modes that can further enhance the nonlinear coupling between the fields. However, the overarching effects that is at play in all types of possible geometries is the resonant aspect of the third order nonlinear coefficient.”

REVIEWER COMMENTS

Reviewer #2 (Remarks to the Author):

I feel that the authors have addressed all comments of the referees and the revised manuscript deserves to be published in Nat Comm

Reviewer #3 (Remarks to the Author):

The authors have satisfactorily addressed my comments. This paper deserves to be published as is.